# A General Large Neighborhood Search Framework for Solving Integer Linear Programs

**Jialin Song**[†]     **Ravi Lanka**[‡] [*]     **Yisong Yue**[†]     **Bistra Dilkina**[§]

[†] California Institute of Technology
[‡] Rakuten, India
[§] University of Southern California

## Abstract

This paper studies a strategy for data-driven algorithm design for large-scale combinatorial optimization problems that can leverage existing state-of-the-art solvers in general purpose ways. The goal is to arrive at new approaches that can reliably outperform existing solvers in wall-clock time. We focus on solving integer linear programs, and ground our approach in the large neighborhood search (LNS) paradigm, which iteratively chooses a subset of variables to optimize while leaving the remainder fixed. The appeal of LNS is that it can easily use any existing solver as a subroutine, and thus can inherit the benefits of carefully engineered heuristic or complete approaches and their software implementations. We show that one can learn a good neighborhood selector using imitation and reinforcement learning techniques. Through an extensive empirical validation in bounded-time optimization, we demonstrate that our LNS framework can significantly outperform compared to state-of-the-art commercial solvers such as Gurobi.

## 1   Introduction

The design of algorithms for solving hard combinatorial optimization problems remains a valuable and challenging task. Practically relevant problems are typically NP-complete or NP-hard. Examples include any kind of search problem through a combinatorial space, such as network designs [19], mechanism design [15], planning [39], inference in graphical models [50], program synthesis [36], verification [7], and engineering design [12, 37], amongst many others.

The widespread importance of solving hard combinatorial optimization problems has spurred intense research in designing approximation algorithms and heuristics for large problem classes, such as integer programming [8, 21, 34] and satisfiability [51, 13, 16]. Historically, the design of such algorithms was done largely manually, requiring careful understandings of the underlying structure within specific classes of optimization problems. Such approaches are often unappealing due to the need to obtain substantial domain knowledge, and one often desires a more automated approach.

In recent years, there has been an increasing interest to automatically learn good (parameters of) algorithms for combinatorial problems from training data. The most popular paradigm, also referred to as "learning to search", aims to augment existing algorithmic templates by replacing hard-coded heuristic components with parameterized learnable versions. For example, this has been done in the context of greedy search for NP-hard graph problems [31]. However, greedy as well as most general purpose heuristic or local search algorithms are limited to combinatorial optimization problems, where the constraints are easy to satisfy, and hence are difficult to apply to domains with intricate side constraints. On the other hand, Integer Linear Programs (ILPs) are a widely applicable problem class that can encode a broad set of domains as well as large number and variety of constrains.

---

[*] Work done while at Jet Propulsion Laboratory, California Institute of Technology

Branch-and-bound, which is a complete search procedure, is the state-of-the-art approach to ILPs and has also been recently extensively researched through the lens of "learning to search" [23, 30, 31, 46, 47, 22]. While this line of research has shown promise, it falls short of delivering practical impact, especially in improving wall-clock time. Practically improving the performance of branch-and-bound algorithms through learning is stymied by the need to either modify commercial solvers with limited access to optimized integration, or to modify open-source solvers such as SCIP [1], which is considerably slower than leading commercial solvers such as Gurobi and CPlex (usually by a factor of 10 or more) [38, 40].

Motivated by the aforementioned drawbacks, we study how to design abstractions of large-scale combinatorial optimization problems that can leverage existing state-of-the-art solvers as a generic black-box subroutine. Our goal is to arrive at new approaches that can reliably outperform leading commercial solvers in wall-clock time, can be applicable to broad class of combinatorial optimization problems, and is amenable to data-driven design. We focus on solving integer linear programs (ILPs), which are a common way to represent many combinatorial optimization problems. We leverage the large neighborhood search (LNS) paradigm [2], an incomplete algorithm that iteratively chooses a subset of variables to optimize while leaving the remainder fixed. A major appeal of LNS is that it can easily use any existing solver as a subroutine, including ones that can handle general ILPs.

Our contributions can be summarized as:

- We propose a general LNS framework for solving large-scale ILPs. Our framework enables easy integration of existing solvers as subroutines, and does not depend on incorporating domain knowledge in order to achieve strong performance. In our experiments, we combine our framework with Gurobi, a leading commercial ILP solver.

- We show that, perhaps surprisingly, even using a *random* decision procedure within our LNS framework significantly outperforms Gurobi on many problem instances.

- We develop a learning-based approach that predicts a partitioning of the integer variables, which then serves as a learned decision procedure within our LNS framework. This procedure is effectively learning how to decompose the original optimization problem into a series of sub-problems that can be solved much more efficiently using existing solvers.

- We perform an extensive empirical validation across several ILP benchmarks, and demonstrate superior wall-clock performance compared to Gurobi across all benchmarks. These results suggest that our LNS framework can effectively leverage leading state-of-the-art solvers to reliably achieve substantial speed-ups in wall-clock time.

## 2    Related Work on Learning to Optimize

An increasingly popular paradigm for the automated design and tuning of solvers is to use learning-based approaches. Broadly speaking, one can categorize most existing "learning to optimize" approaches into three categories: (1) learning search heuristics such as for branch-and-bound; (2) tuning the hyperparameters of existing algorithms; and (3) learning to identify key substructures that an existing solver can exploit. In this section, we survey these paradigms.

**Learning to Search.** In learning to search, one typically operates within the framework of a search heuristic, and trains a local decision policy from training data. Perhaps the most popular search framework for integer programs is branch-and-bound [34], which is a complete algorithm for solving integer programs (ILPs) to optimality. Branch-and-bound is a general framework that includes many decision points that guide the search process, which historically have been designed using carefully attained domain knowledge. To arrive at more automated approaches, a collection of recent works explore learning data-driven models to outperform manually designed heuristics, including learning for branching variable selection [30, 22], or node selection [23, 46, 47]. Moreover, one can also train a model to decide when to run primal heuristics endowed in many ILP solvers [31]. Many of these approaches are trained as policies using reinforcement or imitation learning.

Writing highly optimized software implementations is challenging, and so all previous work on learning to branch-and-bound were implemented within existing software frameworks that admit interfaces for custom functions. The most common choice is the open-source solver SCIP [1], while some previous work relied on callback methods with CPlex [9, 30]. However, in general, one cannot depend on highly optimized solvers being amenable to incorporating learned decision procedures as

subroutines. For instance, Gurobi, the leading commercial ILP solver according to [38, 40], has very limited interface capabilities, and to date, none of the learned branch-and-bound implementations can reliably outperform Gurobi in wall-clock time.

Beyond branch-and-bound, other search frameworks that are amenable to data-driven design include A* search [46], direct forward search [29], and sampling-based planning [11]. These settings are less directly relevant, since our work is grounded in solving ILPs. However, the LNS framework can, in principle, be interpreted more generally to include these other settings as well, which is an interesting direction for future work.

**Algorithm Configuration.** Another area of using learning to speed up optimization solvers is algorithm configuration [26, 28, 4, 5, 33]. Existing solvers tend to have many customizable hyperparameters whose values strongly influence the solver behaviors. Algorithm configuration aims to optimize those parameters on a problem-by-problem basis to speed up the solver.

Similar to our approach, algorithm configuration approaches leverage existing solvers. One key difference is that algorithm configuration does not yield fundamentally new approaches, but rather is a process for tuning the hyperparameters of an existing approach. As a consequence, one limitation of algorithm configuration approaches is that they rely on the underlying solver being able to solve problem instances in a reasonable amount of time, which may not be possible for hard problem instances. Our LNS framework can thus be viewed as a complementary paradigm for leveraging existing solvers. In fact, in our experiments, we perform a simple version of algorithm configuration. We defer incorporating more complex algorithm configuration procedures as future work.

**Learning to Identify Substructures.** The third category of approaches is learning to predict key substructures of an optimization problem. A canonical example is learning to predict backdoor variables [17], which are a set of variables that, once instantiated, the remaining problem simplifies to a tractable form [16]. Our approach bears some high-level affinity to this paradigm, as we effectively aim to learn decompositions of the original problem into a series of smaller subproblems. However, our approach makes a much weaker structural assumption, and thus can more readily leverage a broader suite of existing solvers. Other examples of this general paradigm include learning to pre-condition solvers, such as generating an initial solution to be refined with a downstream solver, which is typically more popular in continuous optimization settings [32].

## 3 A General Large Neighborhood Search Framework for ILPs

We now present our large neighborhood search (LNS) framework for solving integer linear programs (ILPs). LNS is a meta-approach that generalizes neighborhood search for optimization, and iteratively improves an existing solution by local search. As a concept, LNS has been studied for over two decades [45, 2, 41]. However, previous work studied specialized settings with domain-specific decision procedures. For example, in [45], the definition of neighborhoods is highly specific to vehicle routing, and so the decision making of how to navigate the neighborhood is also domain-specific. We instead aim to develop a general framework that avoids requiring domain-specific structures, and whose decision procedures can be designed in a generic and automated way, e.g., via learning as described in Section 4. In particular, our approach can be viewed as a decomposition-based LNS framework that operates on generic ILP representations, as described in Section 3.2.

### 3.1 Background

Formally, let $X$ be the set of all variables in an optimization problem and $\mathcal{S}$ be all possible value assignments of $X$. For a current solution $s \in \mathcal{S}$, a neighborhood function $N(s) \subset \mathcal{S}$ is a collection of candidate solutions to replace $s$, afterwards a solver subroutine is evoked to find the optimal solution within $N(s)$. Traditional neighborhood search approaches define $N(s)$ explicitly, e.g., the 2-opt operation in the traveling salesman problem [18]. LNS defines $N(s)$ implicitly through a *destroy* and a *repair* method. A destroy method destructs part of the current solution while a repair method rebuilds the destroyed solution. The number of candidate repairments is potentially exponential in the size of the neighborhood, which explains the "large" in LNS.

In the context of solving ILPs, the LNS is also used as a local search heuristics for finding high quality incumbent solutions [44, 24, 25]. The ways large neighborhoods are constructed are random [44], manually defined [24] and via bandit algorithm selection from a pre-defined set [25]. Further-

---

**Algorithm 1** `Decomposition-based LNS`

---

1: **Input:** an optimization problem $P$, an initial solutions $S_X$, a decomposition $X = X_1 \cup X_2 \cup \cdots \cup X_k$, a solver $F$
2: **for** $i = 1, \cdots, k$ **do**
3:    $S_X = \text{FIX\_AND\_OPTIMIZE}(P, S_X, X_i, F)$
4: **end for**
5: **return** $S_X$

---

more, these LNS approaches often require interface access to the underlying solver, which is often undesirable when designing frameworks that offer ease of deployment.

Recently, there has been some work on using learning within LNS [27, 49]. These approaches are designed for specific optimization problems, such as capacitated vehicle routing, and so are not directly comparable with our generic approach for solving ILPs. Furthermore, they often focus on learning the underlying solver (rather than rely on existing state-of-the-art solvers), which makes them unappealing from a deployment perspective.

### 3.2 Decomposition-based Large Neighborhood Search for Integer Programs

We now describe the details of our LNS framework. At a high level, our LNS framework operates on an ILP via defining decompositions of its integer variables into disjoint subsets. Afterwards, we can select a subset and use an existing solver to optimize the variables in that subset while holding all other variables fixed. The benefit of this framework is that it is completely generic to any ILP instantiation of any combinatorial optimization problem.

Without loss of generality, we consider the cost minimization objective. We first describe a version of LNS for integer programs based on decompositions of integer variables which is a modified version of the evolutionary approach proposed in [44], outlined in Alg 1. For an integer program $P$ with a set of integer variables $X$, we define a decomposition of the set $X$ as a disjoint union $X_1 \cup X_2 \cup \cdots \cup X_k$. Assume we have an existing feasible solution $S_X$ to $P$, we view each subset $X_i$ of integer variables as a local neighborhood for search. We fix integers in $X \setminus X_i$ with their values in the current solution $S_X$ and optimize for variable in $X_i$ (referred as the `FIX_AND_OPTIMIZE` function in Line 3 of Alg 1). As the resulting optimization is a smaller ILP, we can use any off-the-shelf ILP solver to carry out the local search. In our experiments, we use Gurobi to optimize the sub-ILP. A new solution is obtained and we repeat the process with the remaining subsets.

**Decomposition Decision Procedures.** Notice that a different decomposition defines a different series of LNS problems and the effectiveness of our approach proceeds with a different decomposition for each iteration. The simplest implementation is to use a random decomposition approach, which we show empirically already delivers very strong performance. We can also consider learning-based approaches that learn a decomposition from training data, discussed further in Section 4.

## 4 Learning a Decomposition

In this study, we apply data-driven methods, such as imitation learning and reinforcement learning, to learn policies to generate decompositions for the LNS framework described in Section 3.2. We specialize a Markov decision process for our setting. For a combinatorial optimization problem instance $P$ with a set of integer variables $X$, a state $s \in \mathcal{S}$ is a vector representing an assignment for variables in $X$, i.e., it is an incumbent solution. An action $a \in \mathcal{A}$ is a decomposition of $X$ as described in Section 3.2. After running LNS through neighborhoods defined in $a$, we obtain a (new) solution $s'$. The reward $r(s, a) = J(s) - J(s')$ where $J(s)$ is the objective value of $P$ when $s$ is the solution. We restrict to finite-horizon task of length $T$ so we set the discount factor $\gamma$ to be 1.

### 4.1 Imitation Learning

In imitation learning, demonstrations (from an expert) serves as the learning signals. However, we do not have the access to an expert to generate good decompositions. Instead, we sample random decompositions and take the ones resulting in best objectives as demonstrations. This procedure is shown in Alg 2. The core of the algorithm is shown on Lines 7-12 where we repeatedly sample ran-

---

**Algorithm 2** `COLLECT_DEMOS`

---

1: **Input:** a collection of optimization problems $\{P_i\}_{i=1}^n$ with initial solutions $\{S_i\}_{i=1}^n$, $T$ the number of LNS
    iterations, $m$ the number of random decompositions to sample, $k$ the number of subsets in a decompositon,
    $F$ a solver.
2: **for** $i = 1, \cdots, n$ **do**
3:    $best\_obj \leftarrow \infty$
4:    $best\_decomp \leftarrow None$
5:    **for** $j = 1, \cdots, m$ **do**
6:      $decomps \leftarrow []$
7:      **for** $t = 1, \cdots, T$ **do**
8:        $X \leftarrow$ `RANDOM_DECOMPOSITION`$(P_i, k)$
9:        $S_i \leftarrow$
10:       `Decomposition-based LNS`$(P_i, S_i, X, F)$
11:       $decomps.append(X)$
12:      **end for**
13:      **if** $J(S_i) < best\_obj$ **then**
14:        $best\_obj \leftarrow J(S_i)$
15:        $best\_decomp \leftarrow decomps$
16:      **end if**
17:    **end for**
18:    Record $best\_decomp$ for $P_i$
19: **end for**
20: **return** $best\_decompos$

---

---

**Algorithm 3** `Forward Training for LNS`

---

1: **Input:** a collection of optimization problems $\{P_i\}_{i=1}^n$ with initial solutions $\{S_i\}_{i=1}^n$, $T$ the time horizon,
    $m$ the number of random decompositions to sample, $k$ the number of subsets in a decompositon, $F$ a
    solver..
2: **for** $t = 1, \cdots, T$ **do**
3:    $\{D_i\}_{i=1}^n =$
4:    `COLLECT_DEMOS`$(\{P_i\}_{i=1}^n, \{S_i\}_{i=1}^n, 1, m, k, F)$
5:    $\pi_t =$ `SUPERVISE_TRAIN`$(\{D_i\}_{i=1}^n)$
6:    **for** $i = 1, \cdots, n$ **do**
7:      $X \leftarrow \pi_t(P_i, S_i)$
8:      $S_i \leftarrow$ `Decomposition-based LNS`$(P_i, S_i, X, F)$
9:    **end for**
10: **end for**
11: **return** $\pi_1, \pi_2, \cdots, \pi_T$

---

dom decompositions and call the `Decomposition-based LNS` algorithm (Alg 1) to evaluate
them. In the end, we record the decompositions with the best objective values (Lines 13-16).

Once we have generated a collection of good decompositions $\{D_i\}_{i=1}^n$, we apply two imita-
tion learning algorithms. The first one is behavior cloning [42]. By turning each demon-
stration trajectory $D_i = (s_0, a_0, s_1, a_1, \cdots, s_{T-1}, a_{T-1})$ into a collection of state-action pairs
$\{(s_0, a_0), (s_1, a_1), \cdots, (s_{T-1}, a_{T-1})\}$, we treat policy learning as a supervised learning problem.
In our case, the action $a$ is a decomposition which we represent as a vector. Each element of the
vector indicates which subset $X_i$ this particular variable belongs to. Thus, we reduce the learning
problem to a supervised classification task.

Behavior cloning suffers from cascading errors [43]. We use the forward training algorithm [43]
to correct mistakes made at each step. We adapt the forward training algorithm for our use case
and present it as Alg 3. The main difference with behavior cloning is the adaptive demonstration
collection step on Line 4. In this case, we do not collect all demonstrations beforehand, instead, they
are collected dependent on the predicted decompositions of previous policies.

### 4.2 Reinforcement Learning

For reinforcement learning, for simplicity, we choose to use REINFORCE [48] which is a classical
Monte-Carlo policy gradient method for optimizing policies. The goal is to find a policy $\pi$ that max-

imizes $\eta(\pi) = \mathbb{E}_\pi[\sum_{t=0}^\infty \gamma^t r(s_t, a_t)]$, the expected discounted accumulative reward. The policy $\pi$ is normally parameterized with some $\theta$. Policy gradient methods seek to optimize $\eta(\pi_\theta)$ by updating $\theta$ in the direction of: $\nabla_\theta \eta(\pi_\theta) = \mathbb{E}_{\pi_\theta}[\sum_{t=0}^T \nabla_\theta \log \pi_\theta(a_t|s_t) \sum_{t'=t}^T r(s_{t'}, a_{t'})]$. By sampling trajectories $(s_0, a_0, \cdots, s_{T-1}, a_{T-1}, s_T)$, one can estimate the gradient $\nabla_\theta \eta(\pi_\theta)$.

## 4.3 Featurization of an Optimization Problem

In this section, we describe the featurization of two classes of combinatorial optimization problems.

**Combinatorial Optimization over Graphs.** The first class of problems are defined explicitly over graphs as those considered in [29]. Examples include the minimum vertex cover, the maximum cut and the traveling salesman problems. The (weighted) adjacency matrix of the graph contains all the information to define the optimization problem so we use it as the feature input to a learning model.

**General Integer Linear Programs.** There are other classes of combinatorial optimization problems that do not originate from explicit graphs such as those in combinatorial auctions. Nevertheless, they can be modeled as integer linear programs. We construct the following incidence matrix $A$ between the integer variables and the constraints. For each integer variable $x_i$ and a constraint $c_j$, $A[i, j] = \text{coeff}(x_i, c_j)$ where $\text{coeff}(x_i, c_j)$ is the coefficient of the variable $x_i$ in the constraint $c_j$ if it appears in it and 0 otherwise.

**Incorporating Current Solution.** As outlined in Section 3.2, we seek to adaptively generate decompositions based on the current solution. Thus we need to include the solution in the featurization. Regardless of which featurization we use, the feature matrix has the same number of rows as the number of integer variables we consider, so we can simply append the variable value in the solution as an additional feature.

# 5 Emprical Validation

We present experimental results on four diverse applications covering both combinatorial optimization over graphs and general ILPs. We discuss the design choices of crucial parameters in Section 5.1, and present the main results in Sections 5.2 & 5.3. Finally, we provide an empirical evaluation of a state-of-the-art learning to branch model with Gurobi in Section 5.4 to highlight the necessity of our proposed approach for practical impact.

## 5.1 Datasets & Setup

**Datasets.** We evaluate on 4 NP-hard benchmark problems expressed as ILPs. The first two, minimum vertex cover (MVC) and maximum cut (MAXCUT), are graph optimization problems. For each problem, we consider two random graph distributions, the Erdős-Rényi (ER) [20] and the Barabási-Albert (BA) [3] random graph models. For MVC, we use graphs of size 1000. For MAXCUT, we use graphs of size 500. All the graphs are weighted and each vertex/edge weight is sampled uniformly from [0, 1] for MVC and MAXCUT, respectively. We also apply our method to combinatorial auctions [35] and risk-aware path planning [39], which are not based on graphs. We use the Combinatorial Auction Test Suite (CATS) [35] to generate auction instances from two distributions: regions and arbitrary. For each distribution, we consider two sizes: 2000 items with 4000 bids and 4000 items with 8000 bids. For the risk-aware path planning experiment, we use a custom generator to generate obstacle maps with 30 obstacles and 40 obstacles.

**Learning a Decomposition.** When learning the decomposition, we use 100 instances for training, 10 for validation and 50 for testing. When using reinforcement learning, we sample 5 trajectories for each problem to estimate the policy gradient. For imitation learning based algorithms, we sample 5 random decompositions and use the best one as demonstrations. All our experiment results are averaged over 5 random seeds.

**Initialization.** To run large neighborhood search, we require an initial feasible solution (typically quite far from optimal). For MVC, MAXCUT and CATS, we initialize a feasible solution by including all vertices in the cover set, assigning all vertices in one set and accepting no bids, respectively. For risk-aware path planning, we initialize a feasible solution by running Gurobi for 3 seconds. This time is included when we compare wall-clock time with Gurobi.

|  | MVC BA 1000 | MVC ER 1000 | MAXCUT BA 500 | MAXCUT ER 500 |
|---|---|---|---|---|
| Gurobi | $440.08 \pm 1.26$ | $482.15 \pm 0.82$ | $-3232.53 \pm 16.61$ | $-4918.07 \pm 12.43$ |
| Random-LNS | $433.59 \pm 0.51$ | $471.21 \pm 0.36$ | $\mathbf{-3583.63 \pm 3.81}$ | $-5488.49 \pm 6.60$ |
| BC-LNS | $433.09 \pm 0.53$ | $\mathbf{470.20 \pm 0.34}$ | $\mathbf{-3584.90 \pm 4.02}$ | $\mathbf{-5494.76 \pm 6.51}$ |
| FT-LNS | $\mathbf{432.00 \pm 0.52}$ | $\mathbf{470.04 \pm 0.37}$ | $\mathbf{-3586.29 \pm 3.33}$ | $\mathbf{-5496.29 \pm 6.69}$ |
| RL-LNS | $434.16 \pm 0.38$ | $471.52 \pm 0.15$ | $\mathbf{-3584.70 \pm 1.49}$ | $-5481.57 \pm 2.97$ |

Table 1: Comparison of different LNS methods and Gurobi for MVC and MAXCUT problems.

|  | CATS Regions 2000 | CATS Regions 4000 | CATS Arbitrary 2000 | CATS Arbitrary 4000 |
|---|---|---|---|---|
| Gurobi | $-94559.9 \pm 2640.2$ | $-175772.9 \pm 2247.89$ | $-69644.8 \pm 1796.9$ | $-142168.1 \pm 4610.0$ |
| Random-LNS | $-99570.1 \pm 790.5$ | $-201541.7 \pm 1131.1$ | $-85276.6 \pm 680.9$ | $-170228.3 \pm 1711.7$ |
| BC-LNS | $\mathbf{-101957.5 \pm 752.7}$ | $-207196.2 \pm 1143.8$ | $-86659.6 \pm 720.2$ | $\mathbf{-172268.1 \pm 1594.8}$ |
| FT-LNS | $\mathbf{-102247.9 \pm 709.0}$ | $\mathbf{-208586.3 \pm 1211.7}$ | $-87311.8 \pm 676.0$ | $-169846.7 \pm 5293.2$ |

Table 2: Comparison of different LNS methods and Gurobi for CATS problems.

|  | 30 Obstacles | 40 Obstacles |
|---|---|---|
| Gurobi | $0.7706 \pm 0.23$ | $0.7407 \pm 0.13$ |
| Random-LNS | $0.6487 \pm 0.07$ | $0.3680 \pm 0.03$ |
| BC-LNS | $\mathbf{0.5876 \pm 0.07}$ | $\mathbf{0.3502 \pm 0.07}$ |
| FT-LNS | $\mathbf{0.5883 \pm 0.07}$ | $\mathbf{0.3567 \pm 0.04}$ |

Table 3: Comparison of different LNS methods and Gurobi for risk-aware path planning problems.

|  | MVC BA 1000 | MVC ER 1000 |
|---|---|---|
| Local-ratio | $487.58 \pm 1.16$ | $498.20 \pm 1.24$ |
| Best-LNS | $\mathbf{432.00 \pm 0.52}$ | $\mathbf{470.04 \pm 0.37}$ |

|  | MAXCUT BA 500 | MAXCUT ER 500 |
|---|---|---|
| Greedy | $-3504.79 \pm 7.80$ | $-5302.63 \pm 17.59$ |
| Burer | $\mathbf{-3647.46 \pm 7.63}$ | $\mathbf{-5568.18 \pm 13.47}$ |
| De Sousa | $-3216.86 \pm 9.86$ | $-4994.73 \pm 10.60$ |
| Best-LNS | $-3586.29 \pm 3.33$ | $-5496.29 \pm 6.69$ |

Table 4: (Left) Comparison between LNS with the local-ratio heuristic for MVC. (Right) Comparison between LNS with heuristics for MAXCUT.

|  | CATS Regions 2000 | CATS Regions 4000 | CATS Arbitrary 2000 | CATS Arbitrary 4000 |
|---|---|---|---|---|
| Greedy | $-89281.4 \pm 1296.4$ | $-181003.5 \pm 1627.5$ | $-81588.7 \pm 1657.6$ | $-114015.9 \pm 12313.8$ |
| LP Rounding | $-87029.9 \pm 876.66$ | $-173004.1 \pm 1688.9$ | $-74545.1 \pm 1365.5$ | $-104223.1 \pm 11124.5$ |
| Best-LNS | $\mathbf{-102247.9 \pm 709.0}$ | $\mathbf{-208586.3 \pm 1211.7}$ | $\mathbf{-87311.8 \pm 676.0}$ | $\mathbf{-172268.1 \pm 1594.8}$ |

Table 5: Comparison between LNS with greedy and LP rounding heuristics for CATS.

|  | Set Cover | Maximum Independence Set | Capacitated Facility Location | CATS |
|---|---|---|---|---|
| GCNN [22] | $1489.91 \pm 3.3\%$ | $2024.37 \pm 30.6\%$ | $563.36 \pm 10.7\%$ | $114.16 \pm 10.3\%$ |
| Gurobi | $\mathbf{669.64 \pm 0.55\%}$ | $\mathbf{51.53 \pm 5.25\%}$ | $\mathbf{39.87 \pm 3.91\%}$ | $\mathbf{40.99 \pm 7.19\%}$ |

Table 6: Wall-clock comparison between learning to branch GCNN models [22] and Gurobi.

**Hyperparameter Configuration.** We must set two parameters for our LNS approach. The first is $k$, the number of equally sized subsets to divide variables $X$ into. The second is $t$, how long the solver runs on each sub-problem. A sub-ILP is still fairly large so solving it to optimality can take a long time, so we impose a time limit. We run a parameter sweep over the number of decompositions from 2 to 5 and time limit for sub-ILP from 1 second to 3 seconds. For each configuration of $(k, t)$, the wall-clock time for one iteration of LNS will be different. For a fair comparison, we use the ratio $\Delta/T$ as the selection criterion for the optimal configuration, where $\Delta$ is the objective value improvement and $T$ is the time spent. The configuration results are in Appendix A.1.

## 5.2 Benchmark Comparisons with Gurobi

We now present our main benchmark evaluations. We instantiate our framework in four ways:

- Random-LNS: using random decompositions
- BC-LNS: using a decomposition policy trained using behavior cloning
- FT-LNS: using a decomposition policy trained using forward training
- RL-LNS: using a decomposition policy trained using REINFORCE

We use Gurobi 9.0 as the underlying solver. For learned LNS methods, we sequentially generate 10 decompositions and apply LNS with these decompositions. We use the same time limit setting for running each sub-problem, as a result, the wall-clock among decomposition methods are very close.

We study the incomplete setting, where the goal is to find the best possible feasible solution within a bounded running time. When comparing using just Gurobi, we limit Gurobi's runtime to the longest runtime across all instances from our LNS methods. In other words, Gurobi's runtime is longer than all the decompostion based methods, which gives it more time to find the best solution possible.

**Main Results.** Tables 1, 2 and 3 show the main results. We make two observations:

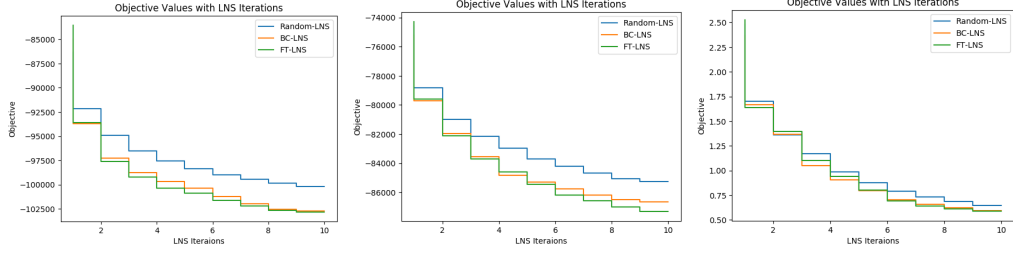

(a) CATS with 2000 items and 4000 bids from regions distribution.

(b) CATS with 2000 items and 4000 bids from arbitrary distribution.

(c) Risk-aware path planning for 30 obstacles.

Figure 1: Improvements of objective values as more iterations of LNS are applied. In all three cases, imitation learning methods, BC-LNS and FT-LNS, outperform the Random-LNS.

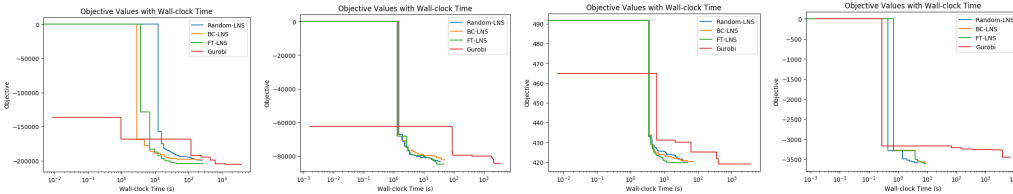

(a) CATS with 4000 items and 8000 bids from regions distribution.

(b) CATS with 2000 items and 4000 bids from arbitrary distribution.

(c) MVC over a Barabási-Albert random graph with 1000 vertices.

(d) MAXCUT over a Barabási-Albert random graph with 500 vertices.

Figure 2: We compare LNS methods on how the objective values improve as more wall-clock time is spent for some representative problem instances. We also include Gurobi in the comparison. All LNS methods find better solutions than Gurobi early on and it takes Gurobi between 2 to 10 times more time to match the solution quality. For MAXCUT (Fig 2d), after running for 2 hours, Gurobi is unable to match the quality of solution found by Random-LNS in 5 seconds.

- All LNS variants significantly outperform Gurobi (up to $50\%$ improvement in objectives), given the same amount or less wall-clock time. Perhaps surprisingly, this phenomenon holds true even for Random-LNS.
- The imitation learning based variants, FT-LNS and BC-LNS, outperform Random-LNS and RL-LNS in most cases.

Overall, these results suggest that our LNS approach can reliably offer substantial improvements over state-of-the-art solvers such as Gurobi. These results also suggest that one can use learning to automatically design strong decomposition approaches, and we provide a preliminary qualitative study of what the policy has learned in Section A.2. It is possible that a more sophisticated RL method could further improve RL-LNS.

**Per-Iteration Comparison.** We use a total of 10 iterations of LNS, and it is natural to ask how the solution quality changes after each iteration. Figure 1 shows objective value progressions of variants of our LNS approach on three datasets. For the two combinatorial auction datasets, BC-LNS and FT-LNS achieve substantial performance gains over Random-LNS after just 2 iterations of LNS, while it takes about 4 for the risk-aware path planning setting. These results show that learning a decomposition method for LNS can establish early advantages over using random decompositions.

**Running Time Comparison.** Our primary benchmark comparison limited all methods to roughly the same time limit. We now investigate how the objective values improve over time by performing 100 iterations of LNS. Figure 2 shows four representative instances. We see that FT-LNS achieves the best performance profile of solution quality vs. wall-clock.

**How Long Does Gurobi Need?** Figure 2 also allows us to compare with the performance profile of Gurobi. In all cases, LNS methods find better objective values than Gurobi early on and maintain this advantage even as Gurobi spends significantly more time. Most notably, in Figure 2d, Gurobi

was given 2 hours of wall-clock time, and failed to match the solution found by Random-LNS in just under 5 seconds (the time axis is in log scale).

### 5.3 Comparison with Domain-Specific Heuristics

We also compare with strong domain-specific heuristics for three classes of problems: MVC, MAX-CUT and CATS. We do not compare in the risk-aware path planning domain, as there are no readily available heuristics. Please refer to Appendix A.4 for descriptions on these heuristics.

Overall, we find that our LNS methods are competitive with specially designed heuristics, and can sometimes substantially outperform them. These results provide evidence that our LNS approach is a promising direction for the automated design of solvers that avoids the need to carefully integrate domain knowledge while achieving competitive or state-of-the-art performance.

Table 4 (Left) summarizes results for MVC. The best LNS (FT-LNS) result outperforms by $11\%$ on BA graphs and $6\%$ on ER graphs. Table 4 (Right) shows results for MAXCUT. The heuristic in Burer et al.[10] performs best for both random graph distributions, which shows that a specially designed heuristic can still outperform a general ILP solver. For CATS, our LNS approach outperforms both heuristics by up to $50\%$ in objective values (Table 5).

### 5.4 Comparison with Learning to Branch Methods

Recently, there has been a surge of interest in applying machine learning methods within a branch-and-bound solver. Most prior work builds on the open-source solver SCIP [1], which is much slower than Gurobi and other commercial solvers [38, 40]. Thus, it is unclear how the demonstrated gains from these methods can translate to wall-clock time improvement. For instance, Table 6 shows a benchmark comparison between Gurobi and a state-of-the-art learning approach built on SCIP [22] (reporting the 1-shifted geometric mean of wall-clock time on hard instances in their paper). These results highlight the large gap between Gurobi and a learning-augmented SCIP, which exposes the issue of designing learning approaches without considering how to integrate with existing state-of-the-art software systems (if the goal is to achieve good wall-clock time performance).

### 5.5 Use SCIP as the ILP Solver

Our LNS framework can incorporate any ILP solver to search for improvement over incumbent solutions. Our main experiments focused on Gurobi because it is a state-of-the-art ILP solver. Here, we also present results on using SCIP as the ILP solver. With the same setting as in Section 5.1 on the CATS Regions distribution with 2000 items and 4000 bids, the results are shown in Table 3. They are consistent with those when Gurobi is used as the ILP solver. Random-LNS significantly outperforms standard SCIP while learning-based methods (BC-LNS and FT-LNS) further improves upon Random-LNS.

|  | CATS Regions 2000 |
| --- | --- |
| SCIP | $-86578.38 \pm 606.21$ |
| Random-LNS | $-98944.90 \pm 645.23$ |
| BC-LNS | $-100513.84 \pm 702.05$ |
| FT-LNS | $-100913.77 \pm 681.00$ |

Figure 3: LNS with SCIP as the ILP solver.

## 6 Conclusion & Future Work

We have presented a general large neighborhood search framework for solving integer linear programs. Our extensive benchmarks show the proposed method consistently outperforms Gurobi in wall-clock time across diverse applications. Our method is also competitive with strong heuristics in specific problem domains. We believe our current research has many exciting future directions that connect different aspects of the learning to optimize research. Our framework relies on a good solver, and thus can be integrated with conventional learning-to-optimize methods. It would also be interesting to study complete search problems, where the goal is to find the globally optimal solution as quickly as possible. We briefly mentioned a version of the algorithm configuration problem we face in Section 5.1, but the full version that adaptively chooses the number of subsets and a time limit is surely more powerful. Finally, our approach is closely tied the problem of identifying useful substructures for optimization. A deeper understanding there can inform the design our data-driven methods and define new learning frameworks, e.g., learning to identify backdoor variables.

# 7 Broad Impact

This paper addresses the general problem of integer linear program optimization which has wide applications in the society. Our contribution allows for faster optimization for these problems. By improving solution qualities within a limited optimization time, one can benefit from, for instance, more efficient resource allocations. In our experiments, we focus on comparing with Gurobi which is a leading commercial integer program solver used by lots of companies. Any concrete improvement is likely to result in positive impact on the day-to-day operations of those companies and the soceity as well.

We see opportunities of research that incorporating existing highly optimized solvers into the machine learning loop beyond integer programs. We encourage research into understanding how to incorporate previous algorithmic developments with novel machine learning components and the benefit of such synergy.

# 8 Acknowledgement

We thank the anonymous reviewers for their suggestions for improvements. Dilkina was supported partially by NSF #1763108, DARPA, DHS Center of Excellence "Critical Infrastructure Resilience Institute", and Microsoft. This research was also supported in part by funding from NSF #1645832, Raytheon, Beyond Limits, and JPL.

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
