[Supplementary Material]

# A   Appendix

## A.1   Algorithm Configuration Results

In this section, we present the algorithm configuration results similar to the one in Section 5.1.

| k \ t | 1 | 2 | 3 |
|---|---|---|---|
| 2 | $13.61 \pm 0.82$ | $14.19 \pm 0.89$ | $\mathbf{14.42 \pm 0.88}$ |
| 3 | $6.06 \pm 0.47$ | $6.17 \pm 0.42$ | $6.65 \pm 0.45$ |
| 4 | $3.09 \pm 0.30$ | $3.14 \pm 0.27$ | $3.61 \pm 0.31$ |
| 5 | $1.84 \pm 0.18$ | $2.13 \pm 0.20$ | $2.08 \pm 0.23$ |

Table 7: Parameter sweep results for $(k, t)$ of an MVC dataset for Erdős-Rényi random graphs with 1000 vertices. Numbers represent improvement ratios $\Delta/t$ for one decomposition, averaged over 5 random seeds.

| k \ t | 1 | 2 | 3 |
|---|---|---|---|
| 2 | $69.05 \pm 2.92$ | $71.87 \pm 2.98$ | $\mathbf{74.14 \pm 3.03}$ |
| 3 | $29.99 \pm 2.07$ | $28.59 \pm 1.80$ | $30.05 \pm 1.81$ |
| 4 | $14.28 \pm 1.04$ | $16.13 \pm 1.13$ | $15.33 \pm 1.19$ |
| 5 | $7.79 \pm 0.77$ | $7.57 \pm 0.72$ | $7.69 \pm 0.69$ |

Table 8: Parameter sweep results for $(k, t)$ of the MVC dataset for Barabási-Albert random graphs with 1000 vertices.

| k \ t | 1 | 2 | 3 |
|---|---|---|---|
| 2 | $2155.60 \pm 22.79$ | $1258.79 \pm 13.65$ | $925.05 \pm 12.50$ |
| 3 | $2700.33 \pm 20.91$ | $1767.37 \pm 10.86$ | $1310.96 \pm 6.25$ |
| 4 | $4454.65 \pm 46.05$ | $4489.60 \pm 49.44$ | $4466.36 \pm 47.20$ |
| 5 | $\mathbf{5414.01 \pm 29,76}$ | $5325.95 \pm 31.07$ | $5404.87 \pm 30.16$ |

Table 9: Parameter sweep results for $(k, t)$ of the MAXCUT dataset for Erdős-Rényi random graphs with 500 vertices.

| k \ t | 1 | 2 | 3 |
|---|---|---|---|
| 2 | $1961.42 \pm 24.54$ | $1043.89 \pm 12.28$ | $1030.60 \pm 2.41$ |
| 3 | $2698.46 \pm 36.44$ | $1887.29 \pm 51.40$ | $1581.43 \pm 55.98$ |
| 4 | $6565.54 \pm 47.36$ | $6454.62 \pm 46.80$ | $\mathbf{6669.28 \pm 47.91}$ |
| 5 | $6400.38 \pm 23.94$ | $6478.23 \pm 19.33$ | $6465.03 \pm 22.54$ |

Table 10: Parameter sweep results for $(k, t)$ of the MAXCUT dataset for Barabási-Albert random graphs with 500 vertices.

| k \ t | 1 | 2 | 3 |
|---|---|---|---|
| 2 | $\mathbf{65360.28 \pm 799.26}$ | $37554.81 \pm 263.48$ | $27864.92 \pm 179.93$ |
| 3 | $61064.41 \pm 519.66$ | $36816.46 \pm 236.11$ | $26633.11 \pm 178.71$ |
| 4 | $56190.18 \pm 530.23$ | $34647.30 \pm 233.18$ | $25547.98 \pm 176.94$ |
| 5 | $54571.21 \pm 344.89$ | $33554.38 \pm 224.77$ | $24238.73 \pm 165.66$ |

Table 11: Parameter sweep results for $(k, t)$ of the CATS dataset for the regions distribution with 2000 items and 4000 bids.

## A.2   Visualization

A natural question is what property a good decomposition has. Here we provide one interpretation for the risk-aware path planning. We use a slightly smaller instance with 20 obstacles for a clearer

| $t$ / $k$ | 1 | 2 | 3 |
|---|---|---|---|
| 2 | **54358.95 ± 1268.30** | 31397.88 ± 364.19 | 21878.70 ± 234.63 |
| 3 | 50046.53 ± 586.72 | 29375.81 ± 336.84 | 20711.09 ± 242.39 |
| 4 | 46449.07 ± 555.02 | 27920.03 ± 315.03 | 20431.02 ± 226.95 |
| 5 | 42190.19 ± 480.57 | 27004.79 ± 315.24 | 19882.16 ± 211.44 |

Table 12: Parameter sweep results for $(k, t)$ of the CATS dataset for the arbitrary distribution with 2000 items and 4000 bids.

| $t$ / $k$ | 1 | 2 | 3 |
|---|---|---|---|
| 2 | 0.37 ± 0.18 | 0.39 ± 0.07 | 0.36 ± 0.05 |
| 3 | 0.41 ± 0.07 | **0.43 ± 0.07** | 0.43 ± 0.07 |
| 4 | 0.37 ± 0.06 | 0.40 ± 0.06 | 0.33 ± 0.05 |
| 5 | 0.33 ± 0.04 | 0.32 ± 0.05 | 0.31 ± 0.05 |

Table 13: Parameter sweep results for $(k, t)$ of the risk-aware path planning for 30 obstacles.

(a) Iteration 2     (b) Iteration 3     (c) Iteration 4     (d) Iteration 5

Figure 4: Visualizing predicted decompositions in a risk-aware path planning problem, with 4 consecutive solutions after 3 iterations of LNS. Each blue square is an obstacle and each cross is a waypoint. The obstacles in red and waypoints in dark blue are the most frequent ones in the subsets that lead to high local improvement.

view. Binary variables in an ILP formulation of this problem model relationships between obstacles and waypoints. Thus we can interpret the neighborhood formed by a subset of binary variables as attention over specific relationships among some obstacles and waypoints.

Figure 4 captures 4 consecutive iterations of LNS with large solution improvements. Each subfigure contains information about the locations of obstacles (light blue squares) and the waypoint locations after the current iteration of LNS. We highlight a subset of 5 obstalces (red circles) and 5 waypoints (dark blue squares) that appear most frequently in the first neighborhood of the current decomposition. Qualitatively, the top 5 obstacles define some important junctions for waypoint updates. For waypoint updates, the highlighted ones tend to have large changes between iterations. Thus, a good decomposition focuses on important decision regions and allows for large updates in these regions.

## A.3 Model Architecture

We first apply PCA to reduce the adjacency matrix obtained in Section 4.3. Then a fully-connected neural network is used to perform the classification task. Table 14 lists the specifications. For MVC problems, for instance, we first apply PCA to reduce the adjacency matrix to 99 dimensions. Then the current solution assignments for each vertex is appended as described in Section 4.3, resulting in a 100-dimensional feature representation for each vertex. Next, it is passed through a hidden layer of 300 units with ReLU activations followed by a 2-class Softmax activations (since the model performs classifications). The number of classes is decided via the hyperparameter search for $k$ as described in Section 5.1 and A.1.

|  | PCA dimensions | Neural Network Architecture | Activation Functions |
|---|---|---|---|
| MVC | 99 | (100, 300, 2) | (ReLU, Softmax) |
| MAXCUT | 299 | (300, 100, 5) | (ReLU, Softmax) |
| CATS 2000 | 99 | (100, 300, 100, 2) | (ReLU, ReLU, Softmax) |
| CATS 4000 | 399 | (400, 300, 2) | (ReLU, Softmax) |
| Path Planning | 499 | (500, 100, 3) | (ReLU, Softmax) |

Table 14: Model architectures for all the experiments.

## A.4 Domain Heuristics

**MVC.** We compare with a 2-OPT heuristic based on local-ratio approximation [6].

**MAXCUT** We compare with 3 heuristics. The first is the greedy algorithm that iteratively moves vertices from one cut set to the other based on whether such a movement can increase the total edge weights. The second, proposed in [10], is based on a rank-two relaxation of an SDP. The third is from [14].

**CATS** We consider 2 heuristics. The first is greedy: at each step, we accept the highest bid among the remaining bids, remove its desired items and eliminate other bids that desire any of the removed items. The second is based on LP rounding: we first solve the LP relaxation of the ILP formulation of a combinatorial auction problem, and then we move from the bid having the largest fractional value in the LP solution down and remove items/bids in the same manner.