[Reviews · NeurIPS 2020]

Review 1

Summary and Contributions: The authors presents a study about applying Large Neighborhood Search (LNS) approach for solving large scale combinatorial problems. Their goal was to out preform the state-of-the-art commercial Gurobi algorithm on wall-clock time.

Strengths: The paper presents the LNS approach and how decompose an optimization problem using the LNS approach given a problem, solver, and a decomposition. They further presents several variants which varied by the decomposing policy. I didn't find a significant novelty in the paper, however, the empirical evaluation does show that even random decomposition surpasses the Gurboi algorithm.

Weaknesses: The paper did not experiment/ elaborate on computational costs difference rather only on wall-clock time.

Correctness: the empirical evaluation seems solid, i would expect to also compare computational costs involved.

Clarity: The paper is clear and well written.

Relation to Prior Work: yes

Reproducibility: Yes

Additional Feedback:


Review 2

Summary and Contributions: The authors propose a large neighborhood search framework for the generally NP-hard Integer Linear Programming (ILP) problems. The tactic employed here is to deploy learning based methods such as reinforcement learning and imitation learning tools to learn a partition of variables for creating sub-problems. Further ILP solvers such as Gurobi can be leveraged as black-box subroutines for solving the sub-problems. Experimental results show that the proposed algorithms outperform the commercial ILP solver Gurobi on several benchmark problems.

Strengths: I enjoy the idea of deploying large neighbor search paradigm for ILP problem solving and it's interesting to see that under the proposed framework even a random strategy for variable partitioning can outperform the Gurobi solver. The experimental results show significant reductions in average running time across the various benchmark problems.

Weaknesses: - The sizes of the benchmark problems do not fully illustrate the practical value of the framework. How is the scalability of this framework when the number of variables increases, say millions, together with a large size of constraints? Since for example, the ILP problems from the structured prediction tasks in computer vision and machine learning are of large sizes. - I wonder if there are some intuitive explanations on the choice of $k$, the number of equally sized subsets. What would be the tradeoff when $k$ is increased and sub-problems with smaller sizes are created, which is more amenable to ILP solvers?

Correctness: The technical details make sense to me.

Clarity: The paper is well written and organized. The experimental results are clearly explained. The Alg 1 presented seems uninformative though. Line 179: Typo "To goal" -> "The goal"

Relation to Prior Work: There are enough discussions on prior work.

Reproducibility: Yes

Additional Feedback:


Review 3

Summary and Contributions: Solving integer linear programs (ILP) is an important field for many real-world optimization problems, such as network design, planning or verification. One approach to solve ILP is to use local search. The authors of the paper at hand propose to learn the decomposition of the search space to speed up large neighborhood search (NLS). To this end, the authors introduce the idea of using imitation learning and reinforcement learning for the learning task. In experiments on several different ILP problem classes, the authors show that their approach can outperform random decompositions effectively.

Strengths: Using imitation learning and RL is a new view on the decomposition problem of NLS and a bit surprisingly one could in fact learn how to do such a decomposition. The results are convincing and show a substantial improvement in runtime.

Weaknesses: Although ILP is not of the typical problems addressed at NeurIPS, there were other papers on similar problems recently published at NeurIPS or similar conferences (e.g., Kleinberg et al. NeurIPS’19, Weisz et al. ICML’18). Therefore, I would consider the paper as in the scope of NeurIPS. Nevertheless, the impact on the NeurIPS community could be considered rather limited; especially since it is applying standard ideas from imitation learning and RL to a new problem. The authors used REINFORCE which is a fairly outdated approach and the authors acknowledge that more modern RL approaches should be compared against to really quantify whether RL is a promising approach. Therefore, the comparison between imitation learning and RL is not very insightful. The authors argue in length that their approach is ILP-solver-agnostic, which is important since one cannot do easy modification in commercial solvers. Although I agree with the statement, I was hoping to see results with more than one solver (here Gurobi) to show that the approach is indeed effective also with other solvers (e.g., SCIP or CPLEX). That’s even more so since the comparison in Section 5.4 would be more convincing if apples were compared with apples.

Correctness: I have no concerns regarding correctness. The idea and how the ideas are applied are correct as far as I can say. Nevertheless, I raise some open questions in the "additional feedback" field.

Clarity: Although the paper is well written, many important details and algorithm outlines are hidden in the appendix, e.g., Algorithm 2 and Algorithm 3. Fully understanding the approach without reading the appendix is nearly impossible. Since the authors focus anyway more on imitation learning in the experiments, the paper might be cleaner and easier to read if the RL approach is not discussed at all. Surprisingly, the most trivial algorithm (Algorithm 1) is shown in the main paper and the main part (i.e., decomposition) is hidden in the input of Algo 1. Therefore, I don’t know what I learn from Algorithm 1.

Relation to Prior Work: I’m not aware of any missing related work.

Reproducibility: No

Additional Feedback: I wonder whether the used LNS requires a local search algorithm for solving the subproblem (Line 3). Please clarify in rebuttal. The authors argue that they set \gamma to 1 because it is a finite-horizon task. I completely agree that this is a possible choice; however even for finite-horizon tasks, \gamma can be set to values smaller than 1.0. I wonder how sensitive their approach is to such hyperparameters. The authors sampled 5 trajectories for each problem (instance`?) to estimate the policy gradient. I’m not sure whether I understood that point fully. Does that mean for in each iteration of REINFORCE, the current policy is applied 5 times to each instance (100 in training)? This would be crazy expensive. Which instances were used for the hyperparameter configuration optimization? I hope that these were the training instances only. Since Gurobi is a commercial software, the reproducibility of the results is unfortunately limited. >>>>>>>>>>>>>>>>>>>>>>>>>>>>>>> Comments after Rebuttal >>>>>>>>>>>>>>>>>>>>>>>>>>>>>> Thank you very much for the rebuttal. It clarified some of my doubts. In particular, thank you for adding the comparison to SCIP. I have to admit that I don't have much background in combinations of ILP and DL. Therefore, it is hard for me to judge the novelty of the paper. Nevertheless, I have the impression that the paper is more or less only combining two existing ideas in a very straight forward way. However, the results proof that this is indeed a good idea. I wonder how the paper will look like after it is more focused on imitation learning and less on RL. Therefore, I hesitate to increase my score too much.

[Author Response · NeurIPS 2020]

We thank the reviewers for their valuable comments. All three reviews appreciated the significant reduction in wall-clock
time compared with Gurobi in solving several benchmark ILP problems.

**General Comment on Novelty.** We want to first globally address the concern on novelty in the paper. Identifying novel
applications & capabilities of machine learning has long been valued at top machine learning venues. Recent examples
include: [Wei et al., 2020] that optimizes proximal solvers using standard RL approaches (and won outstanding paper
at ICML 2020); [Balunovic et al., 2018] that optimizes SMT solvers using DAgger; and [Gasse et al., 2019] learns
branching decisions in ILP solvers using behavior cloning and graph NNs. In terms of empirical significance, our
work is the first to significantly outperform a state-of-the-art commercial solver such as Gurobi in wall-clock for
general ILP problems, and we do so by identifying the large neighborhood search framework as a suitable one for
incorporating learning. As a result, we believe our work has identified a novel application, learning decompositions
for large neighborhood search (LNS), and obtained convincing empirical results to be highly relevant to the NeurIPS
community. Next, we address individual comments from each reviewer.

**R2. Computational costs.** At test time, all the experiments were carried out on the same hardware with 16 logical-core
Intel(R) Xeon(R) CPU E5-2637 v4 @ 3.50GHz processor and 132 GB of RAM. At training time, RL costs more to
train than IL, while random does not need training. One is normally open to spending training time to obtain improved
test time performance, as shown by improvements of IL over random.

**R3. Scalability.** This is an important question to answer for practical relevance. However, the number of integer
variables and constraints is only a rough measure on how hard an ILP instance is [Van Roy and Wolsey, 1987]. Within
the benchmark problems we considered, the sizes are already difficult as indicated by long running time of Gurobi.
Furthermore, the considered sizes are comparable and, in some cases, exceeding those in recent learning-augmented
ILP solver papers [Gasse et al., 2019, He et al., 2014]. A related scalability issue concerns how feasible it is to learn
decompositions of millions of variables into thousands of subsets. We believe our current method has limitations at such
scales, and studying extensions of our approach (e.g., hierarchical imitation learning) is an interesting future direction.

**R3. Choice of $k$.** For larger $k$, each sub-problem becomes easier to solve, at the expense of smaller neighborhoods, thus
reducing the opportunities to find better solutions per iteration. So it is a trade-off of finding out the largest sub-problem
that is still amenable to ILP solvers while allowing for the maximal neighborhood space for solution improvement.

**R4. Novelty.** Please see the general comment above. Our strong empirical results showcase the value of large
neighborhood search as a framework/application for incorporating learning.

**R4. Choice of RL algorithm.** In retrospect, the inclusion of REINFORCE did not convey much information as our
emphasis was on imitation learning approaches since they were more effective. This discovery is consistent with other
recent works on speeding up ILP solvers [He et al., 2014, Gasse et al., 2019] which employed imitation learning. We
included REINFORCE for completeness and will improve the writing to focus more on imitation learning.

**R4. SCIP.** We thank the reviewer for the suggestion. We ran some experiments on using SCIP as the base solver for
the same combinatorial auction instances from regions on 2000 items and 4000 bids. The results are consistent with
those using Gurobi: LNS methods outperform SCIP and learning delivers
further improvements. The reason we focused on Gurobi in the paper is
because it is by far the fastest ILP solver and we were excited by convinc-
ingly outperforming it with a general framework. We hope the results on
SCIP can convince the reviewer that our method is indeed solver agnostic.
We are happy to include a full suite of experiments on SCIP in the final version of the paper.

| | |
|---|---|
| SCIP | $-86578.38 \pm 606.21$ |
| Random-LNS | $-98944.90 \pm 645.23$ |
| BC-LNS | $-100513.84 \pm 702.05$ |
| FT-LNS | $-100913.77 \pm 681.00$ |

**R4. Addtional feedback. Solve sub-problem:** we use a solver, e.g., Gurobi or SCIP, to solve the sub-problem, which
is an ILP as well. **REINFOCE samples:** you are correct – it is very computationally expensive, which is another
reason we decided to focus on imitation learning. **Tuning parameters:** we used 50 training instances.

# References

M. Balunovic, P. Bielik, and M. Vechev. Learning to solve smt formulas. In *NeurIPS*, 2018.

M. Gasse, D. Chételat, N. Ferroni, L. Charlin, and A. Lodi. Exact combinatorial optimization with graph convolutional neural
networks. In *NeurIPS*, 2019.

H. He, H. Daume III, and J. M. Eisner. Learning to search in branch and bound algorithms. In *NeurIPS*, 2014.

T. J. Van Roy and L. A. Wolsey. Solving mixed integer programming problems using automatic reformulation. *Operations Research*,
35(1):45–57, 1987.

K. Wei, A. Aviles-Rivero, J. Liang, Y. Fu, C.-B. Schnlieb, and H. Huang. Tuning-free plug-and-play proximal algorithm for inverse
imaging problems. In *ICML*, 2020.


[Meta-Review · NeurIPS 2020]

This paper received positive reviews from all three reviewers but during the discussion there was widespread concern about whether the contribution is of sufficient significance for a NeurIPS publication. In particular, the question was raised whether a paper that merely applies ML techniques in a new application domain was of sufficient significance. I also read the paper and the author's rebuttal and I very much agree with the authors on this point: application papers have always been a part of the major ML conferences and can help drive the field forward. I am therefore happy to recommend acceptance and encourage the authors to spend more text in the final version towards motivating the problem to a general audience.